# Do Judge an Entity by its Name!
# Entity Typing using Language Models

Russa Biswas[1,2], Radina Sofronova[1,2], Mehwish Alam[1,2],
Nicolas Heist[3], Heiko Paulheim[3], and Harald Sack[1,2]

[1] FIZ Karlsruhe – Leibniz Institute for Information Infrastructure, Germany
`firstname.lastname@fiz-karlsruhe.de`,
[2] Karlsruhe Institute of Technology, Institute AIFB, Germany
[3] University of Mannheim, Germany
`{nico,heiko}@informatik.uni-mannheim.de`

**Abstract.** The entity type information in a Knowledge Graph (KG) plays an important role in a wide range of applications in Natural Language Processing such as entity linking, question answering, relation extraction, etc. However, the available entity types are often noisy and incomplete. Entity Typing is a non-trivial task if enough information is not available for the entities in a KG. In this work, neural language models and a character embedding model are exploited to predict the type of an entity from only the name of the entity without any other information from the KG. The model has been successfully evaluated on a benchmark dataset.

**Keywords:** Entity Type Prediction · Knowledge Graph Completion · Deep Neural Networks.

## 1 Introduction

Entity Typing is a vital task in Knowledge Graph (KG) completion and construction. The entity types in KGs such as DBpedia, YAGO, Wikidata, etc. are either extracted automatically from structured data, generated using heuristics, or are human-curated. These factors lead to incomplete and noisy entity type information in the KGs. More specifically, in case of DBpedia, the Wikipedia infoboxes are the primary source of information. The types of the entities in Wikipedia infoboxes are mapped to the classes in DBpedia. Recent years have witnessed research in the automated prediction of entity types in KGs using heuristics [5] as well as neural network-based models [1, 3, 4]. The existing state-of-the-art (SOTA) models exploit the triples in the KGs whereas others consider the textual entity descriptions as well. While those approaches work well if there is a lot of information about an entity, it is still a challenge to type entities for which there is only scarce information. This paper focuses on predicting the entity types solely from their label names, e.g., *Is it possible to predict that the entity* `dbr:Berlin` *is a place only from its name?*. To do so, the SOTA continuous space-based Neural Language Models (NLM) such as Word2Vec, GloVe,

Wikipedia2Vec [11], BERT [6] as well as a character embedding model are exploited. This work tackles the challenge of insufficient information for the entities. Since the NLMs are trained on a huge amount of textual data, they provide implicit contextual information about the entities in their corresponding latent representations. In this work, the task of entity typing is considered as a classification problem in which a neural network-based classifier is applied on top of the NLMs. Furthermore, an analysis of the performance of the different NLMs for this task is provided.

## 2   Related Work

A heuristic based approach SDType [5] leverages the relations between the instances to predict the types of the entities. In [3, 4], the authors propose embedding based entity typing models considering the structural information in the KG as well as the textual entity descriptions. The word embedding models such as Word2Vec, GloVe, FastText are trained on KGs in [1] to generate the entity vectors to predict the types of entities. Other language model based entity typing models are proposed in MuLR [10] and FIGMENT [9] in which multi-level representations of entities are learned by using character, word, and entity embeddings. However, these entity type prediction models based on NLMs do not restrict themselves to only the label names and consider the other information available in the KGs. In [8], the authors propose a model in which the pre-trained RDF2Vec vectors are used to predict the entity types using a classifier. Also, the meaningfulness of the entity names in Semantic Web has been studied in [7]. However, unlike the SOTA models, in this work, the NLMs are leveraged to generate the entity embeddings from the names of the entities for the task of entity type prediction.

## 3   Model

This section discusses the NLMs and the classifiers used for the task of entity typing only from the names of the entities.

***Word2Vec.*** It aims to learn the distributed representation for words reducing the high dimensional word representations in a large corpus. The CBOW Word2Vec model predicts the current word from a window of context words and the skip-gram model predicts the context words based on the current word.

***GloVe.*** GloVe exploits the global word-word co-occurrence statistics in the corpus with the underlying intuition that the ratios of word-word co-occurrence probabilities encode some form of the meaning of the words.

***BERT.*** **B**idirectional **E**ncoder **R**epresentations from **T**ransformers is a contextual information based embedding approach in which pretraining on bidirectional representations from unlabeled text by using the left and the right context in all the layers is performed.

***Wikipedia2vec.*** The model jointly learns word and entity embeddings from Wikipedia where similar words and entities are close to one another in the vector

space. It uses three submodels to learn the representation namely: Wikipedia Link Graph Model, Word-based skip-gram model, and Anchor context model.

**Character Embedding.** Character embedding represents the latent representations of characters trained over a corpus which helps in determining the vector representations of out-of-vocabulary words.

**Embeddings of the Entity Names.** In this work, pre-trained Word2Vec model on Google News dataset[4], GloVe model pre-trained on Wikipedia 2014 version and Gigaword 5[5], Wikipedia2Vec model pre-trained on English Wikipedia 2018 version[6], and pre-trained English character embeddings derived from GloVe 840B/300D dataset[7] are used with a vector dimension of 300. The average of all word vectors in the entity names is taken as the vector representation of the entities. For BERT, the average of the last four hidden layers of the model is taken as a representation of the names of entities and the dimension used is 768.

**Classification.** In this work, entity typing is considered a classification task with the types of entities as classes. Two classifiers have been built on top of the NLMs: (**i**) Fully Connected Neural Network (FCNN), and (**ii**) Convolutional Neural Network (CNN). A three-layered FCNN model consisting of two dense layers with ReLU as an activation function has been used on the top of the vectors generated from the NLMs. The softmax function is used in the last layer to calculate the probability of the entities belonging to different classes. The CNN model consists of two 1-D convolutional layers followed by a global max-pooling layer. `ReLu` is used as an activation function in the convolutional layers and the output of the pooling layer is then passed through a fully connected final layer, in which the softmax function predicts the classes of the entities.

## 4  Evaluation

This section consists of a detailed description of the datasets used for evaluating the models, followed by an analysis of the results obtained.

**Datasets.** The experiments are conducted on the benchmark dataset DBpedia630k [12] extracted from DBpedia consisting of 14 non-overlapping classes[8] with 560,000 train and 70,000 test entities. However, predicting fine-grained type information of an entity only from its name is a non-trivial task. For e.g. identifying `dbr:Kate_Winslet` as an *Athlete* or *Artist* from only the entity name is challenging. Therefore, seven coarse-grained classes of the entities in this dataset are considered: *dbo:Organisation*, *dbo:Person*, *dbo:MeanOfTransportation*, *dbo:Place*, *dbo:Animal*, *dbo:Plant*, and *dbo:Work*. Also, 4.656% of the total entities in the train set and 4.614% entities in the test set have their type information mentioned in their RDF(S) labels. For example, *dbr:Cybersoft_(video_game_company)* has the label *Cybersoft (video game company)* stating that it is a *Company*.

---

[4] https://code.google.com/archive/p/word2vec/
[5] http://nlp.stanford.edu/data/glove.6B.zip
[6] https://wikipedia2vec.github.io/wikipedia2vec/pretrained/
[7] https://github.com/minimaxir/char-embeddings/blob/master/output/
[8] https://bit.ly/3bBgjiV

**Table 1.** Results on the DBpedia630k dataset (in accuracy %)

| Embedding Models | Types in Labels | | no Types in Labels | | CaLiGraph Test Set | |
|---|---|---|---|---|---|---|
| | FCNN | CNN | FCNN | CNN | FCNN | CNN |
| word2vec | 80.11 | 46.71 | 72.08 | 44.39 | 48.93 | 25.91 |
| GloVe | 83.34 | 54.06 | 82.62 | 53.41 | 61.88 | 31.3 |
| wikipedia2vec | 91.14 | 60.47 | 90.68 | 57.36 | 75.21 | 36.97 |
| BERT | 67.37 | 62.27 | 64.63 | 60.4 | 53.42 | 35.55 |
| character embedding | 73.43 | 58.13 | 72.66 | 58.3 | 54.91 | 45.73 |

Therefore, the experiments are conducted both with and without the type information in the names for the DBpedia630k dataset. To evaluate the approaches independently of DBpedia, we use an additional test set[9] composed of entities from CaLiGraph [2]. The latter is a Wikipedia-based KG containing entities extracted from tables and enumerations in Wikipedia articles. It consists of 70,000 entities that are unknown to DBpedia and evenly distributed among 7 classes.

***Results.*** The results in Table 1 depict that for all the NLMs, FCNN works better compared to the CNN model. This is because the CNN model does not work well in finding patterns in the label names of the entities. Also, BERT performs the worst in predicting the type of the entities from their label names. Further error analysis shows that only 4.2% of the total person entities in the test set *with Types in Labels* variation of the dataset have been correctly identified as `dbo:Person` for BERT. Since the names of persons can be ambiguous and BERT is a contextual embedding model, the vector representations of the entities generated only from their label names do not provide a proper latent representation of the entity. However, FCNN achieves an accuracy of 84.74% on the same dataset without the class `dbo:Person` for BERT. On the other hand, Wikipedia2Vec works best amongst all the NLMs for FCNN with an accuracy of 91.14% and 90.68% on the *Types in Labels* and *no Types in Labels* variants of the dataset respectively. Also, on removal of the class `dbo:Person` from the dataset, it achieves an accuracy of 91.01% on *Types in Labels* variant. Therefore, the decrease of 0.13% in the accuracy infers that entities of the class `dbo:Person` are well represented in the entity vectors obtained from the pre-trained Wikipedia2Vec model.

However, after removing the type information from the name labels, a slight drop in the accuracy for each model has been observed for both the classifiers. Wikipedia2Vec and the character embedding model experience the smallest drop in accuracy of 0.46% and 0.77% with the FCNN classifier. This is because DBpedia entities are extracted from Wikipedia articles, therefore the vectors of the entities are well represented by the Wikipedia2Vec model. Also for character embedding, removal of the type information from their labels has low impact because the vector representation of the entity names depends on the corresponding character vectors and not word vectors. Furthermore, an unseen test

---

[9] http://data.dws.informatik.uni-mannheim.de/CaLiGraph/whats-in-a-name/whats-in-a-name_caligraph_test-balanced70k.csv.bz2

set from CaLiGraph has been evaluated on the classification model trained on the *no Types in Labels* variation of the dataset. On the CaLiGraph test set, the FCNN model achieves the best results with the Wikipedia2Vec model with an accuracy of 75.21%. The entities in the CaLiGraph test set are not contained in DBpedia, hence the representations of these entities are not learned during the training of the Wikipedia2Vec model. This depicts the robustness of the proposed model and the entity vectors generated by taking average of the word vectors present in the names of the entities provides a better latent representation.

## 5 Conclusion and Future Work

In this paper, different NLMs for entity typing in a KG have been analyzed. The achieved results imply that NLMs can be exploited to get enough information to predict the types of entities in a KG only from their names. In the future, fine-grained type prediction using other features from the KG using the NLMs is to be explored.

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
