# OpenReview forum: "Do Judge an Entity by its Name! Entity Typing using Language Models"
_eswc-conferences.org/ESWC/2021/Conference/Poster_and_Demo_Track — ESWC2021 P&D_

### Official Review · AnonReviewer4 · 2021-04-07
**Not very significant but ok for a poster paper**

**Rating:** 6
**Confidence:** 3

**Review:**

This poster paper presents an approach, based on neural networks applied on language models, to predict the type of an entity from its name only. It is well written. The originality of the paper comes from the fact that existing approaches to predict the type of an entity use more than the name, usually triples from knowledge graphs or textual descriptions. However, the proposed approach is not very original: the authors use existing language models (embedding models) and existing machine learning techniques. Experiments show the accuracy on 5 embedding models used with 2 neural networks. 3 datasets are used: 2 based on labels from DBpedia resources (one with the standard label, one where the eventual type information is removed from the label), and the last one (CaLiGraph) is composed of entities from Wikipedia that are not in DBpedia. Using a set of out-of-base entities is a good idea but more information would be welcome. On each embedding model, how many entities from CaLiGraph are not represented in the model?


Pros
-	Really simple input: the name of an entity. This could be helpful for many tasks that need entity typing and have no more information than a name.
-	Some experiments are conducted.
-	The results are analyzed.

Cons
-	Types chosen in the experiments are really coarse-grained. It would be interesting to see up to what kind of granularity the classifier still performs well.
-	I would have liked to see a comparison of the results with a baseline using more than the name. Would the results be significantly different?
-	I wonder what the results would be if we give the entity names to a human being and ask him/her to make the classification. What would be the accuracy? Better or worse than your approach?

In conclusion, the contribution is not very significant but fair enough for a poster. Therefore, I accept the paper.

**Anonymity:**

Yes, I would like my review to remain anonymous.

---

### Official Review · AnonReviewer1 · 2021-04-14
**This paper studies the entity typing problem based on entity names using language models.**

**Rating:** 5
**Confidence:** 4

**Review:**

This paper studies the problem of entity typing. It classifies entity types only based on entity names. Thus, it resorts to language models to enhance the information needed. Based on Table 1, it seems that wikipedia2vec performs much better. But is it because the data leakage problem? Furthermore, why do you only consider FCNN and CNN in comparison?

**Anonymity:**

Yes, I would like my review to remain anonymous.

---

### Official Review · AnonReviewer2 · 2021-04-16
**Entities and their name for typing. A Poster**

**Rating:** 8
**Confidence:** 4

**Review:**

Entity typing is both challenging and very useful for NLP and KGs, and often insufficient information is available. This Poster paper shows, however, that very often sufficient social semantics is encoded in the names of the entities that their type can be predicted with rather high precision.

To do this, the authors apply several SOTA NLmethods to predict types from names with relatively high performance, and compare the different methods on a DBPedia based testset.

There are, of course, a number of shortcomings for this paper, e.g. the rather limitted discussion of related work, the dependence on a dataset that concentrates on very popular instances, which will have been close to the datasets on which the prediction models were trained, etc.

But this paper is well written, describes a nice initial idea which is well-executed, so for me an ideal poster for ISWC.

**Anonymity:**

Yes, I would like my review to remain anonymous.

---

### Decision · Program_Chairs · 2021-04-19

Accept